# Protective Effects of Collagen Tripeptides in Human Aortic Endothelial Cells by Restoring ROS-Induced Transcriptional Repression

**DOI:** 10.3390/nu13072226

**Published:** 2021-06-29

**Authors:** Hidehito Saito-Takatsuji, Yasuo Yoshitomi, Yasuhito Ishigaki, Shoko Yamamoto, Noriaki Numata, Yasuo Sakai, Masayoshi Takeuchi, Naohisa Tomosugi, Shogo Katsuda, Hideto Yonekura, Takayuki Ikeda

**Affiliations:** 1Department of Biochemistry, Kanazawa Medical University School of Medicine, 1-1 Daigaku, Uchinada, Kahoku-gun, Ishikawa 920-0293, Japan; saitoh@kanazawa-med.ac.jp (H.S.-T.); yositomi@kanazawa-med.ac.jp (Y.Y.); yonekura@kanazawa-med.ac.jp (H.Y.); 2Division of Molecular Oncology and Virology, Department of Life Science, Medical Research Institute, Kanazawa Medical University, 1-1 Daigaku, Uchinada, Kahoku, Ishikawa 920-0293, Japan; ishigaki@kanazawa-med.ac.jp; 3Technical Center, Jellice Co., Ltd., 4-4-1 Sakae, Tagajo, Miyagi 985-0833, Japan; s.yamamoto@jellice.com (S.Y.); numata@jellice.com (N.N.); sakai@jellice.com (Y.S.); 4Division of AGEs Research, Department of Advanced Medicine, Medical Research Institute, Kanazawa Medical University, 1-1 Daigaku, Uchinada, Kahoku, Ishikawa 920-0293, Japan; takeuchi@kanazawa-med.ac.jp; 5Division of Aging Research, Department of Advanced Medicine, Medical Research Institute, Kanazawa Medical University, 1-1 Daigaku, Uchinada, Kahoku, Ishikawa 920-0293, Japan; tomosugi@kanazawa-med.ac.jp; 6Department of Pathology II, Kanazawa Medical University School of Medicine, 1-1 Daigaku, Uchinada, Kahoku-gun, Ishikawa 920-0293, Japan; katuda4297@gmail.com

**Keywords:** collagen tripeptide, oxidative stress, human aortic endothelial cells, gene expression

## Abstract

Collagen tripeptide (CTP) is defined as a functional food material derived from collagenase digests of type I collagen and contains a high concentration of tripeptides with a Gly-X-Y sequence. CTP has several biological effects, including the acceleration of fracture healing, ameliorating osteoarthritis, and improving dryness and photoaging of the skin. Recently, an antiatherosclerotic effect of CTP has been reported, although its molecular mechanism is yet to be determined. In this study, we examined the effects of CTP on primary cultured human aortic endothelial cells (HAECs) under oxidative stress, because oxidative endothelial dysfunction is a trigger of atherosclerosis. DNA microarray and RT-qPCR analyses showed that CTP treatment recovered the downregulated expression of several genes, including the interleukin-3 receptor subunit alpha (*IL3RA*), which were suppressed by reactive oxygen species (ROS) treatment in HAECs. Furthermore, IL3RA knockdown significantly decreased the viability of HAECs compared with control cells. RT-qPCR analysis also showed that solute carrier 15 family peptide transporters, which are involved in CTP absorption into cells, were expressed in HAECs at levels more than comparable to those of a CTP-responsive human osteoblastic cell line. These results indicated that CTP exerts a protective effect for HAECs, at least in part, by regulating the recovery of ROS-induced transcriptional repression.

## 1. Introduction

Collagen tripeptide (CTP) is a functional food material prepared from porcine type I collagen by digestion [1]. CTP is a highly purified, nonantigenic, and low-allergenic tripeptide fraction of collagen digests and contains tripeptides with Gly-X-Y sequences [1,2]. CTP is efficiently absorbed and detected in blood rapidly after CTP ingestion [3]. Furthermore, the tripeptide component of CTP is selectively absorbed into connective tissue according to whole-body autoradiography with the administration of a single dose of tritium-labeled Gly-Pro-Hyp [4].

A number of studies have already revealed the biological effects of CTP. Tsuruoka et al. demonstrated that the oral administration of CTP by rats with femur fractures accelerated fracture healing by promoting type I collagen gene expression [2], whereas another study reported that periodic knee injections of CTP delayed cartilage degeneration in an experimental osteoarthritis rabbit model [5]. Additionally, Hakuta et al. reported an anti-inflammatory effect of CTP for patients with atopic dermatitis and inhibitory effects of secretions of inflammatory cytokines and chemokines in vitro [6]. Additionally, oral administration of CTP improved skin barrier function in ultraviolet B-exposed hairless mice [7] and dryness or pruritus of dry skin in humans [8,9]. For atherosclerosis, an inhibitory effect of CTP on atherosclerosis development was shown in atherosclerosis model rabbits [10]. Recently, Tomosugi et al. have reported that CTP may have contributed to the prevention and treatment of atherosclerosis in healthy humans [11]. However, the molecular mechanisms underlying the protective effects of CTP for atherosclerosis are yet to be determined.

The formation of atherosclerotic plaques is a complex process that includes endothelial dysfunction, neovascularization, apoptosis, matrix degradation, inflammation, and thrombosis [12,13]. Endothelial dysfunction has been identified as an early predictor of atherosclerosis and cardiovascular disease, and oxidative stress has a pivotal role in atherosclerosis progression [13,14,15]. In addition, vascular endothelial cells (ECs) line the inner wall of the vasculature and have several important functions, including vascular tone, the regulation of immune cell migration, and platelet activity. These findings indicate that the dysfunction of ECs is a trigger to atherosclerosis progression. It is thus important to protect ECs from oxidative stress to prevent the development of atherosclerosis.

To identify the mechanisms underlying the protective effects of CTP on atherosclerosis, this study examined the effects of CTP on global gene expression changes in human aortic ECs (HAECs) under oxidative stress and showed that CTP exerts a protective effect on HAECs through the recovery of reactive oxygen species (ROS)-induced transcriptional repression.

## 2. Materials and Methods

### 2.1. Materials

CTP containing more than 50% tripeptide components (CTP-50) and more than 90% tripeptide components (Tp-100) were provided by Jellice Co., Ltd. (Miyagi, Japan). HuMedia-EG2 medium containing fetal bovine serum (FBS), hydrocortisone hemisuccinate, human epidermal growth factor (hEGF), human fibroblastic growth factor basic (hFGF-B), and heparin was purchased from Kurabo Corp. (Osaka, Japan). Hydrogen peroxide (H_2_O_2_) was obtained from Wako (Osaka, Japan), whereas oligonucleotides were acquired from Eurofins Genomics (Tokyo, Japan).

### 2.2. Cell Culture

Primary cultured HAECs and primary cultured human aortic smooth muscle cells (SMCs) were purchased from LifeLine^®^ Cell Technology (Frederick, MD, USA). Cells at one passage after thawing the original frozen cell stock were used for all experiments. HAECs were cultured at 37 °C at 5% CO_2_ and maintained in HuMedia-EG2 medium containing 2% (*v*/*v*) FBS, 10 ng/mL hEGF, 1.34 µg/mL hydrocortisone hemisuccinate, and 10 µg/mL heparin (growth medium) according to the manufacturer’s instructions (Kurabo Corp.). SMCs were cultured at 37 °C at 5% CO_2_ and maintained in HuMedia-SG2 medium containing 5% (*v*/*v*) FBS, 5 ng/mL hEGF, 5 ng/mL hFGF-B, and 5 µg/mL insulin according to the manufacturer’s instructions (Kurabo Corp.). Human osteoblastic hFOB1.19 cells were purchased from the American Tissue Culture Collection (Manassas, VA, USA) and maintained in DMEM/F-12 medium containing 10% (*v*/*v*) FBS, 15 mM HEPES, and 0.3 mg/mL G418, as described previously [2]. HEK293TN cells were purchased from System Biosciences (Palo Alto, CA, USA) and maintained in DMEM (Wako) supplemented with 10% FBS and 100 IU/mL penicillin/100 µg/mL streptomycin.

### 2.3. DNA Microarray Analysis

HAECs were seeded at 5000 cells/cm^2^ and cultured in growth medium. At 60% cell density, cells were preincubated in HuMedia-EG2 medium containing 1% (*v*/*v*) FBS, 5 ng/mL hFGF-B, and 10 µg/mL heparin (assay medium) at 37 °C for 2 h. After a 2 h preincubation, H_2_O_2_ (200 µM) was added, and cells were then incubated at 37 °C for 10 min. Next, CTP (CTP-50) was added to the culture at 100 µg/mL and incubated at 37 °C for 24 h. After CTP treatment, total RNA was extracted from untreated HAECs, H_2_O_2_-treated HAECs, and H_2_O_2_- and CTP-treated HAECs using an RNeasy Micro Kit (Qiagen, Hilden, Germany). RNA quality was determined by the ratio of 28S to 18S ribosomal RNA band intensities on electrophoretic gels under denaturing conditions. Subsequently, 300 ng of total RNA was labeled using a GeneChip WT Sense Target Labeling Kit (Affymetrix, Santa Clara, CA, USA) according to the manufacturer’s instructions. Fragmented and labeled cDNAs were then hybridized onto Affymetrix GeneChip Human Gene 1.0 ST arrays in a GeneChip Hybridization Oven 640 (Affymetrix). Arrays were then washed and stained using a GeneChip Fluidics Station 450 and detected using a 3000 7G GeneChip Scanner (Affymetrix). All arrays passed the quality-control criteria of Expression Console (Affymetrix). Raw data CEL files were then normalized using the RMA algorithm, and data were exported using Expression Console or Gene Spring (Agilent Technologies, Santa Clara, CA, USA). DNA microarray analyses of untreated SMCs, H_2_O_2_-treated SMCs, and H_2_O_2_- and CTP-treated SMCs was performed identically to HAECs cultured in HuMedia-SG2 medium containing 2.5% (*v*/*v*) FBS.

### 2.4. Quantitative Real-Time Reverse Transcription-Polymerase Chain Reaction (RT-qPCR) Analysis

HAECs were seeded at 5000 cells/cm^2^ in 6-well plates and cultured in growth media. At 60% cell density, cells were preincubated in the assay medium at 37 °C for 2 h. After preincubation, H_2_O_2_ (200 µM) was added, and cells were incubated at 37 °C for 10 min. CTP (Tp-100) was then added to the culture at 100 µg/mL and incubated at 37 °C for 4 h. The same volume of phosphate-buffered saline (PBS) was added as a control. After incubation, total RNA was isolated from the cells using an RNeasy Micro Kit. Gene expression was analyzed via RT-qPCR, using a StepOnePlus Real-Time PCR System (Applied Biosystems, Foster City, CA, USA). The primers used for RT-qPCR are provided in Table 1. RT-qPCR was performed using 100 ng total RNA and a One-step SYBR^®^ PrimeScript™ PLUS RT-PCR Kit (TaKaRa, Otsu, Japan) in a final reaction volume of 20 μL. Reaction mixtures were incubated at 42 °C for 5 min for reverse transcription and then at 95 °C for 10 s, followed by 40 cycles of 95 °C for 5 s and 60 °C for 30 s. After detection using a StepOnePlus Real-Time PCR System, data were analyzed via the average Ct value and normalized to that of an internal standard (β-actin).

### 2.5. Production of Lentiviral Vector Encoding shRNA Sequences against Interleukin-3 Receptor Subunit Alpha (IL3RA)

shRNA vectors were constructed using a pSIH-H1-copGFP shRNA vector (System Biosciences) by inserting synthetic oligonucleotides. The following four unique shRNA constructs were then prepared using synthetic oligonucleotides: (1) 5′-AGAGACAGAACCTCCTTCCAGCTACTCAA-3′; (2) 5′-AGGAGTGTCTGGTGACTGAAGTACAGGTC-3′; (3) 5′-AGGCGTCAACAGTACGAGTGTCTTCACTA-3′; and (4) 5′-GTGCGGAGAATCTGACCTGCTGGATTCAT-3′. Pseudoviral particles were generated by cotransfecting a pPACKH1 Packaging Plasmid Mix (System Biosciences) and a mixture of the four pSIH vectors into HEK293TN cells in a 6-well plate using Lipofectamine and Plus Reagent (Life Technologies, Carlsbad, CA, USA), as described previously [16,17]. For controls, a pSIH-H1-copGFP vector inserted with a scrambled sequence (5′-TAGCGACTAAACACATCAA-3′) that had no homology with any human mRNA (shCont) and no vector [Lenti (−)] was used. Supernatants were collected after 72 h from the cultured medium and used for infection.

### 2.6. Transduction of Lentiviral Vector

HAECs were seeded at a density of 8000 cells/cm^2^ in 24-well plates and incubated at 37 °C for 12 h, followed by the addition of lentivirus-containing medium at a 2:1 ratio of lentivirus-containing medium to growth medium and incubated for 48 h. After 48 h incubation, total RNA was isolated from untreated cells [Lenti (−)], control shRNA lentivirus-infected cells (shCont), and IL3RA shRNA lentivirus-infected cells (shIL3RA) using an RNeasy Micro Kit (Qiagen). The medium of lentivirus-infected cells was then replaced with a fresh medium at a 1:1 ratio of growth medium to assay medium and further incubated for 48 h at 37 °C for a total incubation duration of 96 h. At 96 h after infection, cell viability was assayed by the dye exclusion method using trypan blue.

### 2.7. Statistical Analysis

The statistical significance of the data was determined using the Student’s t-test for unpaired data. Multiple comparisons were analyzed via one-way ANOVA followed by Tukey’s honestly significant difference test. A *p*-value < 0.05 was considered statistically significant.

## 3. Results

### 3.1. Microarray Analysis of Gene Expression Changes under Oxidative Stress and Effects of CTP in HAECs and SMCs

We examined the effects of CTP on the overall gene expression in HAECs under oxidative stress with H_2_O_2_ [18]. First, the concentration-dependent effects of H_2_O_2_ on HAECs were checked. HAECs were treated with various concentrations of H_2_O_2_ (0, 100, 200, 500, and 1000 µM) for 24 h, and they were then analyzed for cell morphology and viability. As shown in Figure 1A, cells treated with 100 μM H_2_O_2_ exhibited no morphological changes. At a concentration of 200 µM, cells showed a slightly elongated morphology, and fewer rounded and detached cells were detected, but most cells were still alive. In contrast, cells treated with H_2_O_2_ at a concentration of 500 µM or higher became rounded and detached from the plate (Figure 1A). Thus, the treatment condition of 200 µM was adopted for DNA microarray analysis. CTP treatment did not affect the cell morphology of either control cells (data not shown) or H_2_O_2_-treated cells (Figure 1A).

For Affymetrix microarray analysis, HAECs were cultured for 24 h with 100 µg/mL CTP after treatment with 200 µM H_2_O_2_ for 10 min. Total RNA was isolated from untreated HAECs (control), HAECs treated with H_2_O_2_, and HAECs treated with H_2_O_2_ and CTP (H_2_O_2_ + CTP), followed by DNA microarray analysis. Based on DNA microarray analysis, we detected 201 probe sets that were downregulated by less than 0.5-fold in H_2_O_2_-treated cells compared with control cells (Figure 1B), and only seven probe sets that were upregulated more than twofold in H_2_O_2_-treated cells compared with control cells (Figure 1B). Thus, most differentially expressed genes in H_2_O_2_-treated cells were found to be downregulated. It is implicit that genes downregulated by H_2_O_2_ which are recovered by CTP treatment likely play vital roles in the effects of CTP on HAECs. By comparing H_2_O_2_-treated cells with H_2_O_2_ + CTP-treated cells (Figure 1C), 13 probe sets (nine genes; Table 2) that were upregulated more than 1.5-fold in H_2_O_2_ + CTP-treated cells were detected among those genes that were downregulated less than 0.5-fold in H_2_O_2_-treated cells. Notably, CTP treatment had no significant effects on the global gene expression of untreated cells (data not shown).

For the microarray analysis of SMCs, SMCs were cultured and treated with H_2_O_2_ and CTP, similar to HAECs. SMCs treated with H_2_O_2_ at 200 µM had no morphological changes, with the appearance of only a few rounded and detached cells compared to control cells, and CTP treatment did not affect the cell morphology of H_2_O_2_-treated cells (Figure 1D). Based on DNA microarray analysis, 32 probe sets that were downregulated less than 0.5-fold in H_2_O_2_-treated cells compared with control cells were detected (Figure 1E). Additionally, 20 probe sets that were upregulated more than twofold in H_2_O_2_-treated cells compared with control cells were detected (Figure 1E). These results may suggest that HAECs are more sensitive to oxidative stress than SMCs. Furthermore, no gene that was upregulated more than 1.5-fold in H_2_O_2_ + CTP-treated cells was detected among those genes that were downregulated less than 0.5-fold in H_2_O_2_-treated cells (Figure 1F). Moreover, the global transcriptomes of the untreated, H_2_O_2_-treated, and H_2_O_2_ + CTP-treated HAECs and SMCs were compared using PCA. The result showed that the H_2_O_2_-induced gene expression changes in HAECs were partially restored to the original gene expression by CTP treatment, whereas no such tendency was observed in SMCs (Figure 1G). Thus, SMCs were not further examined; instead, this study focused on analyzing the functions of the candidate genes identified in HAECs.

### 3.2. RT-qPCR Analysis of Genes Recovered in CTP-Treated HAECs

To confirm the microarray results, RT-qPCR was performed on different batches of RNA samples from independent experiments. Among the nine genes that were downregulated in H_2_O_2_-treated cells and recovered in H_2_O_2_ + CTP-treated cells, the *IL3RA* gene, which encodes the alpha subunit of IL3R, was chosen [19], and its expression was analyzed by RT-qPCR, because it is expressed and has been determined to be vital in vascular ECs [20,21,22]. HAECs were treated with or without H_2_O_2_ (200 µM) for 10 min and further cultured for 4 h in the presence or absence of 100 µg/mL CTP. After total RNA extraction, IL3RA expression was analyzed using the β-actin gene as an endogenous control. We confirmed that IL3RA mRNA levels were significantly decreased in H_2_O_2_-treated cells compared with control cells and increased in H_2_O_2_ + CTP-treated cells compared with H_2_O_2_-treated cells (Figure 2A). Importantly, CTP treatment had no significant effects on IL3RA expression in control cells (Figure 2A), suggesting that CTP has few adverse effects on healthy cells. These results indicated that CTP treatment recovers IL3RA expression downregulated by H_2_O_2_ in HAECs.

### 3.3. Impacts of IL3RA Suppression on HAECs

*IL3RA* encodes the alpha subunit of heterodimeric IL3R (IL3Rα), which determines the receptor binding specificity for IL-3 [19]. IL3RA is expressed and is vital in vascular ECs [20,21,22]; therefore, the impacts of IL3RA suppression in HAECs were examined. For this analysis, HAECs were infected with a lentivirus that expressed an shRNA to IL3RA mRNA. Transduction efficiency was confirmed at 48 h after infection by GFP, which was expressed from the lentivirus vector backbone. Most cells expressed GFP in lentivirus-infected cells (shIL3RA; Figure 2B), indicating efficient lentivirus infection in HAECs. Furthermore, there was no difference in the number of cells between IL3RA shRNA lentivirus-infected and control cells at this time point (Figure 2B). Using total RNA isolated from uninfected cells [Lenti (−)], control shRNA lentivirus-infected cells (shCont), and IL3RA shRNA lentivirus-infected cells (shIL3RA), IL3RA mRNA levels at this time point were examined by RT-qPCR. As shown in Figure 2C, the mRNA level of IL3RA in IL3RA knockdown cells was decreased to less than 10% of Lenti (−) and control cells. Lentivirus-infected cells were further incubated for an additional 48 h for a total duration of 96 h, and cell morphology and viability were then observed. As shown in Figure 2D, IL3RA knockdown resulted in cells becoming rounded and detached from the plate surface after 96 h of transduction. Additionally, the cell viability of IL3RA knockdown cells was significantly lower than control cells (Figure 2E). These results indicated that IL3RA suppression induces cell death in HAECs.

VEGF is a well-known essential autocrine survival factor in ECs. Hypoxia induces VEGF expression to EC, stimulating proliferation and survival [23]. VEGF is also a key factor required for tumor progression and survival to circumvent their hypoxic environment [24,25,26]. Thus, we next examined VEGF expression in uninfected cells, control cells, and IL3RA knockdown cells by RT-qPCR. As shown in Figure 2F, the VEGF mRNA levels in both control and IL3RA knockdown cells were upregulated compared with uninfected cells. The VEGF mRNA level in IL3RA knockdown cells was higher than that in control cells (Figure 2F), whereas the cell viability of IL3RA knockdown cells was lower than that in control cells (Figure 2D,E). These results suggested that the suppression of IL3RA signaling pathways may lead to a decrease in cell viability even if VEGF expression is elevated.

### 3.4. Peptide Transporter Expression in HAECs and SMCs

Dipeptides and tripeptides are absorbed directly via the peptide transporter solute carrier 15A1 (SLC15A1) [27], which belongs to the SLC15 family of membrane transporters [28]. Therefore, orally administered CTP should be absorbed into intestinal cells via members of the SLC15 family, and plasma CTP should also be incorporated into cells via the SLC15 family of transporters [4,6]. Thus, we examined the expression of the members of the SLC15 family in HAECs and SMCs and compared them with those found in cells of the human osteoblastic cell line hFOB1.19, which is known to be responsive to CTP [2]. The SLC15 family of membrane transporters consists of four members, i.e., SLC15A1, SLC15A2, SLC15A3, and SLC15A4, which are also known as PEPT1, PEPT2, PHT2, and PHT1, respectively [28]. The expressions of these four SLC15 family members in HAECs and SMCs were analyzed by RT-qPCR using the specific primer pairs listed in Table 1. As shown in Figure 3A, SLC15A1, SLC15A2 and SLC15A4 mRNAs were expressed in HAEC more highly than those in hFOB1.19 cells. In contrast, the mRNA level of SLC15A3 was expressed in HAECs at a lower level than that in hFOB1.19 cells (Figure 3A). These results indicated that HAECs express members of the SLC15 membrane transporter family at levels greater than hFOB1.19 cells, which respond to CTP, and can thus take up CTP across the plasma membrane. In SMCs, SLC15A1 and SLC15A3 mRNA levels were much lower than those in hFOB1.19 cells, whereas SLC15A2 and SLC15A4 mRNA levels were expressed in SMCs more highly than those in hFOB1.19 cells (Figure 3A). SLC15A1 is assumed to be the major transporter for CTP [4,6]; therefore, it is the reason why no gene upregulated more than 1.5-fold in H_2_O_2_ + CTP-treated SMCs was detected among the genes that were downregulated less than 0.5-fold in H_2_O_2_-treated SMCs. We have also analyzed gene expression changes of SLC15 family members in H_2_O_2_-treated HAECs and H_2_O_2_ + CTP-treated HAECs compared with control HAECs using microarray data. As shown in Figure 3B, no expression change was found in any of the four members of the SLC15 family in each condition.

## 4. Discussion

Many studies have revealed the biological effects of CTP, including the acceleration of fracture healing [2], improving experimental osteoarthritis [5], exerting an anti-inflammatory effect for patients with atopic dermatitis [6] and antiphotoaging effects [7], and improving dry skin [8,9]. Previous studies have shown that CTP induces collagen expression via a p38 mitogen-activated protein kinase cascade [29] and inhibits dipeptidyl peptidase-4 activity [30], but the molecular mechanisms and signaling pathways underlying these biological effects have not been fully elucidated.

Recently, we demonstrated that the oral administration of CTP has preventive effects on atherosclerosis in rabbits [10] and humans [11]. In this study, to elucidate the molecular mechanisms underlying the antiatherosclerotic effects of CTP, we examined the effects of CTP on primary cultured HAECs under oxidative stress because oxidative endothelial dysfunction is a trigger of atherosclerosis [13,14,15]. DNA microarray and RT-qPCR analyses showed that CTP treatment recovered *IL3RA* expression, which was suppressed by H_2_O_2_ treatment in HAECs (Figure 1 and Figure 2). RT-qPCR, using different batches of RNA samples from independent experiments, confirmed that suppressed IL3RA expression was recovered by CTP treatment (Figure 2A).

IL3R is described as a heterodimer that shares a beta subunit in common with the receptors for IL-5 and granulocyte-macrophage colony-stimulating factor, and the alpha subunit of IL3R, which is encoded by the *IL3RA* gene, determines the receptor binding specificity for IL-3 [19]. IL3RA is expressed in vascular ECs and has essential roles in controlling human EC tube morphogenesis [20], EC-pericyte tube coassembly [21], and the regulation of lymphatic EC differentiation from blood vascular ECs [22]. In addition, IL-3 has important roles in vascular ECs as a survival or proliferation factor [31] and acts as an endothelial differentiation factor for hematopoietic-derived endothelial progenitor cells [32]. IL-3 has also been reported to stimulate human EC–pericyte tube network coassembly [21,33]. These findings indicate that the IL-3/IL3Rα signaling pathway is essential for vascular endothelial survival and function.

In this study, IL3RA expression was determined to be downregulated by H_2_O_2_, and IL3RA knockdown in HAECs resulted in decreased cell viability compared with control cells (Figure 2). These results indicated that oxidative stress causes the suppression of genes including *IL3RA*, which are involved in the survival of HAECs, and CTP recovers some of the gene suppression and protects HAECs from oxidative dysfunction and cell death. In fact, 91 probe sets were detected, including the probe sets shown in Table 2, which were upregulated more than 1.2-fold in H_2_O_2_ + CTP-treated HAECs among the 201 probe sets downregulated less than 0.5-fold in H_2_O_2_-treated HAECs. Therefore, we posit that the protection of HAECs by CTP is exerted as an additive effect of the recovery of the expression of multiple genes.

Nucleotide-binding oligomerization domain-like receptor family pyrin domain containing 3 (NLRP3) inflammasome is involved in vascular diseases including atherosclerosis [34]. The NLRP3 inflammasome cleaves and activates caspase-1. Active caspase-1 cleaves pro-IL-1β into its active form, IL-1β, inducing inflammatory responses. Active caspase-1 also cleaves gasdermin D (GSDMD) to execute a particular form of programmed cell death called pyroptosis. The N-terminal domain of GSDMD forms pores in the plasma membrane, causing cell lysis and inflammatory responses by releasing IL-1β. IL-1β activates inflammatory signaling, including nuclear factor-κB, in ECs, resulting in the transcriptional activation of NLRP3 and pro-IL-1β to accelerate inflammatory responses. ROS are reported to have an important role in NLRP3 inflammasome activation [35]. It is likely that NLRP3 inflammasome is activated in H_2_O_2_ treatment and alleviated by CTP in this study. Although we analyzed gene expression changes of NLRP3 and pro-IL-1β using microarray data, no notable changes in their expression were found. Therefore, NLRP3 inflammasome may be irrelevant to ROS-induced transcriptional repression and protective effects of CTP for HAECs.

This study also demonstrated that CTP has a restorative effects of oxidative stress-induced gene suppression, but the site of action of CTP and the signaling pathways stimulating gene expression remain unknown. There may be cell surface receptors for CTP from which intercellular signals are generated, but there are currently no insights regarding such cell surface receptors. CTP is directly absorbed into cells via peptide transporters [4,6]; therefore, we examined the expression of the SLC15 family of membrane transporters [28] in HAECs and compared them with those found in cells of the human osteoblastic cell line hFOB1.19, which is known to be responsive to CTP [2]. RT-qPCR analysis showed that all four members of the SLC15 family were expressed in HAECs at levels more than comparable to those of hFOB1.19 cells (Figure 3), indicating that HAECs incorporate CTP through the cell membrane. In addition, intracellular CTP or dipeptides derived from CTP can stimulate gene expression through an intracellular peptide sensor protein, such as a mechanistic target of rapamycin kinase [36], which senses amino acids and short peptides and further stimulates signaling pathways related to transcription, translation, and protein degradation. Finally, it may be stated that the protective effects of CTP against atherosclerosis might be attributed to its action on ECs, because the expression level of SLC15A1, which is the major transporter of CTP, is low in SMCs (Figure 3A), and SMCs are barely responsive to CTP.

## 5. Conclusions

This study revealed that CTP treatment recovered the expression of several genes including IL3RA suppressed by oxidative stress in HAECs. CTP may protect vascular ECs, at least in part, by restoring ROS-induced transcriptional repression, thereby conferring protection from the development of atherosclerosis and other vascular dysfunctions. Further extensive studies are needed to better clarify the molecular mechanisms involved in the antiatherosclerotic effects of CTP and underlying signaling pathways for the effects of CTP. Nevertheless, the present findings have revealed new biological effects of CTP in vascular cells, providing new clues to clarify the pathogenesis of atherosclerosis and other vascular dysfunctions and to develop preventive measures against them.

## Figures and Tables

**Figure 1 nutrients-13-02226-f001:**
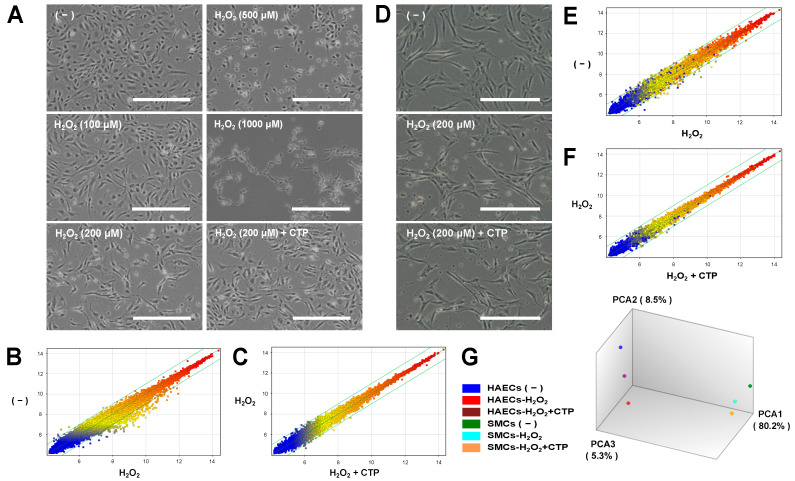
Gene expression analysis of HAECs and SMCs under oxidative conditions in the presence or absence of CTP. (**A**) Effects of H_2_O_2_ on HAECs. HAECs were cultured in various concentrations of H_2_O_2_ (0, 100, 200, 500, and 1000 µM) for 24 h. Representative phase contrast images of HAECs treated with H_2_O_2_ and H_2_O_2_ + CTP. Scale bars: 100 µm. (**B**) Gene expression profiling of H_2_O_2_-treated HAECs (H_2_O_2_) compared with untreated control cells (−). After incubation in the presence or absence of 200 μM H_2_O_2_ for 10 min, HAECs were cultured for 24 h, and DNA microarray analysis was performed. (**C**) Gene expression profiling of CTP- and H_2_O_2_-treated HAECs (H_2_O_2_ + CTP) compared with H_2_O_2_-treated cells (H_2_O_2_). After treatment with 200 μM H_2_O_2_ for 10 min, HAECs were cultured for 24 h with or without 100 μg/mL CTP, and DNA microarray analysis was performed. (**D**) Representative phase contrast images of SMCs treated with H_2_O_2_ and H_2_O_2_ + CTP. Scale bars: 100 µm. (**E**) Gene expression profiling of H_2_O_2_-treated SMCs (H_2_O_2_) compared with untreated control cells (−). After incubation in the presence or absence of 200 μM H_2_O_2_ for 10 min, SMCs were cultured for 24 h, and DNA microarray analysis was performed. (**F**) Gene expression profiling of CTP- and H_2_O_2_-treated SMCs (H_2_O_2_ + CTP) compared with H_2_O_2_-treated cells (H_2_O_2_). After treatment with 200 μM H_2_O_2_ for 10 min, SMCs were cultured for 24 h with or without 100 μg/mL CTP, and DNA microarray analysis was performed. (**G**) Global transcriptome analyses of the untreated, H_2_O_2_-treared, and H_2_O_2_ + CTP-treated HAECs and SMCs using a principal component analysis (PCA). The expression pattern of H_2_O_2_ + CTP-treated HAECs moved in a direction closer to that of untreated HAECs. In contrast, the expression pattern of H_2_O_2_ + CTP-treated SMCs did not move like HAECs.

**Figure 2 nutrients-13-02226-f002:**
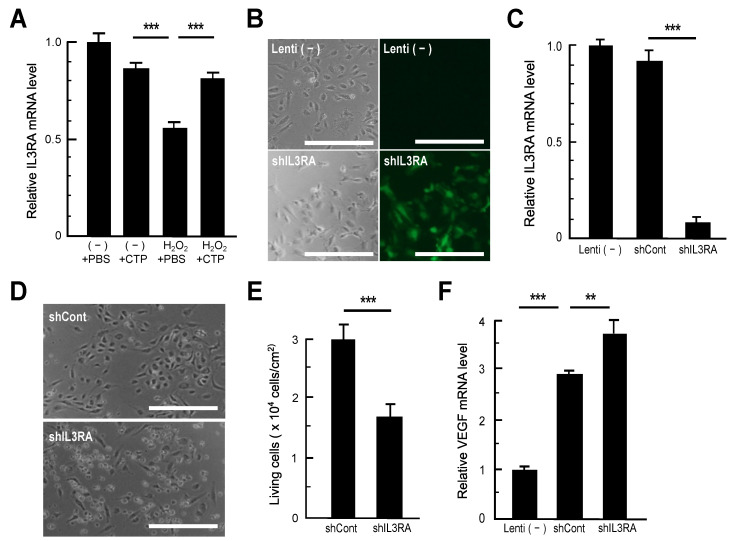
The impact of IL3RA suppression on HAECs. (**A**) RT-qPCR confirmed that IL3RA expression was suppressed in H_2_O_2_-treated HAECs and restored in H_2_O_2_ + CTP-treated HAECs. After 2 h preincubation in the assay medium, HAECs in 6-well plates were incubated with 200 μM H_2_O_2_ for 10 min. CTP (100 µg/mL) was then added to the culture and incubated for 4 h. The same volume of PBS was added as a control. Total RNA was isolated and analyzed by RT-qPCR. Values were compared with the values of untreated control cells [(−) + PBS]. Data are expressed as the mean ± SEM (n = 3). Comparative results were obtained from three independent experiments. *** *p* < 0.001. (**B**) Lentivirus transduction efficiency in HAECs. HAECs in 24-well plates were incubated in IL3RA shRNA lentivirus- or control shRNA lentivirus-containing medium for 48 h. At this time point, GFP expression in lentivirus-infected cells was observed under fluorescence microscopy, confirming that most cells expressed GFP. Representative phase contrasts and fluorescence images of uninfected [Lenti (−)] and IL3RA knockdown (shIL3RA) cells are shown. Scale bars: 100 µm. (**C**) IL3RA mRNA expression was analyzed by RT-qPCR. Values were compared with the values of uninfected cells [Lenti (−)]. The IL3RA mRNA level in IL3RA knockdown cells was decreased to less than 10% of uninfected and control cells. Data are expressed as the mean ± SEM (n = 3). *** *p* < 0.001. (**D**) Effects of IL3RA knockdown on HAECs. Cultures of lentivirus-infected HAECs were replaced with a fresh growth medium and incubated for 96 h. Representative phase contrast images of HAECs at 96 h after lentivirus infection. Scale bars: 100 µm. (**E**) Cell viability of IL3RA knockdown cells. Cell viability was assayed by the dye exclusion method with trypan blue. Data are expressed as the mean ± SEM (n = 8). Comparative results were obtained from two independent experiments. *** *p* < 0.001. (**F**) Vascular endothelial growth factor (VEGF) mRNA level in lentivirus-uninfected [Lenti (−)], control lentivirus-infected (shCont), and IL3RA shRNA lentivirus-infected (shIL3RA) HAECs at 48 h after infection. Values were compared with the values of [Lenti (−)]. Data are expressed as the mean ± SEM (n = 3). ** *p* < 0.01, *** *p* < 0.001.

**Figure 3 nutrients-13-02226-f003:**
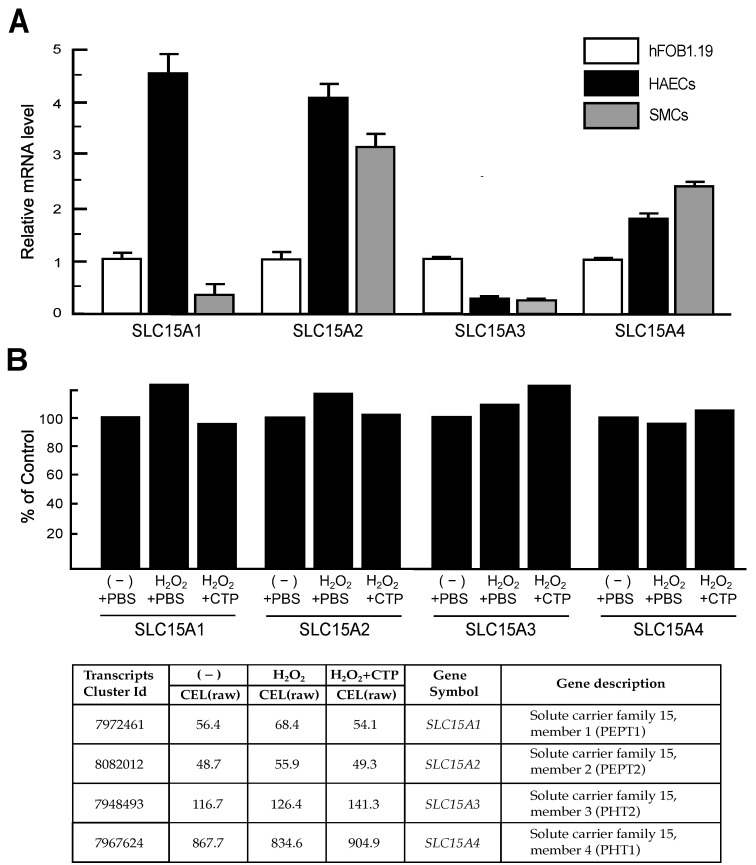
mRNA levels of SLC15 family transporters in HAECs and SMCs. (**A**) Total RNA was isolated, and SLC15A1, SLC15A2, SLC15A3, and SLC15A4 mRNA levels were analyzed via RT-qPCR using the specific primers listed in Table 1. Values were normalized with β-actin mRNA and compared to the values of cells from the known CTP-responsive human osteoblastic cell line hFOB1.19. Comparative results were obtained from three independent experiments and conventional RT-PCR analysis. (**B**) DNA microarray data for the expression of SLC15 family members in control HAECs [(−) + PBS], H_2_O_2_-treated HAECs (H_2_O_2_ + PBS), and H_2_O_2_ + CTP-treated HAECs (H_2_O_2_ + CTP). Bar graphs were created from DNA microarray data, and values were compared with those of the control.

**Table 1 nutrients-13-02226-t001:** Primers used for qPCR and RT-PCR.

Gene	Forward Primer	Reverse Primer
IL3RA	5′-CAGGCGTCAACAGTACGAGT-3′	5′-TTGCAGTCATGTTGGGTGGA-3′
VEGF-A	5′-GCCTCCGAAACATGAACTTTCTGCTG-3′	5′-TGGTGATGTTGGACTCCTCA-3′
SLC15A1 (PEPT1)	5′-ATTCCGCCACAATGTCAACC-3′	5′-CCGTGACAGAGAAGACCACT-3′
SLC15A2 (PEPT2)	5′-CCCTGAAGGAAACATAGTGGCTC-3′	5′-TAGCCAGTGCTGTCGCTTTGGA-3′
SLC15A3 (PHT2)	5′-GCAAGAGGACATCGCCAACTTC-3′	5′-TTGGGATGTGGAGGTGAAGACC-3′
SLC15A4 (PHT1)	5′-CCTCTGAAGGACAAACTGGTCG-3′	5′-ACAAGGTTCAGCCTTTTACTCTCC-3′

**Table 2 nutrients-13-02226-t002:** Genes downregulated less than 0.5-fold in H_2_O_2_-treated cells and upregulated more than 1.5-fold in H_2_O_2_ + CTP-treated HAECs.

Transcript Cluster ID		CEL (Raw)		Fold Change by CTP	Gene Symbol	Gene Description
	(−)	H_2_O_2_	H_2_O_2_ + CTP			
7896750 ^1^	144.8	44.2	94.9	2.15		
8165663 ^1^	204.4	56.2	125	2.22		
8086538	160.9	72.7	155.7	2.14	*LIMD1-AS1*	LIMD1 antisense RNA 1
8113356 ^1^	100.6	46.3	93.2	2.02		
8059852	318.1	154.3	279.3	1.81	*MSL3P1*	Male-specific lethal 3 homolog (Drosophila) pseudogene 1
8176323	246	119.2	213.1	1.79	*IL3RA*	Interleukin 3 receptor, alpha (low affinity)
8097058	322.2	158.8	284	1.79	*CEP170L*	Centrosomal protein 170 kDa-like
8165752	246.2	119.6	213.4	1.78	*IL3RA*	Interleukin 3 receptor, alpha (low affinity)
7916898	407.7	81.1	132.1	1.63	*DEPDC1*	DEP domain containing 1
8165672 ^1^	466.3	158.9	255.7	1.61		
7914878	276.8	74.2	117.5	1.58	*CLSPN*	Claspin
7971653	225.1	80.1	126.3	1.58	*DLEU2*	Deleted in lymphocytic leukemia 2 (non-protein coding)
7979307	936.5	242.2	373.1	1.54	*DLGAP5*	Discs, large (Drosophila) homolog-associated protein 5

^1^ There is no description of gene symbols and gene descriptions for transcript cluster IDs in the present probe set database for the Human Gene 1.0 ST array on the Affymetrix NetAffx website.

## Data Availability

The microarray data presented in this study are openly available in the Gene Expression Omnibus (GEO) under accession number GSE175612.

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
