# Peer review of "Protective Effects of Collagen Tripeptides in Human Aortic Endothelial Cells by Restoring ROS-Induced Transcriptional Repression"

_nutrients, 2021, doi:10.3390/nu13072226_

Round 1

Reviewer 1 Report

Main points:
-Nutrients is a very good quality journal with a good impact factor (4.546). I do not find the quantity of results, although interesting, sufficient. I suggest that the authors study the inflammasome in the absence and presence of PTC under the conditions defined in their experimental design. 

In my opinion, this work is incomplete and needs to be completed by additional results, particularly on the inflammatory component, which is very important in the pathophysiology of atherosclerosis.

-Figure 1 panels B, C, E and F: axes are illegible

Reviewer 2 Report

It is a topical area and the study design exhibit scientific soundness.

Line 300: The author must add 2-3 sentences explaining effect of hypoxia on VEGF expression in EC and cancer cells. The author should include following references

1) Brown JM. Tumor hypoxia in cancer therapy. Methods Enzymol. 2007;435:297-321.2) Kaur H, Li JJ, Bay BH, Yung LY. Investigating the antiproliferative activity of high affinity DNA aptamer on cancer cells. PLoS One. 2013;8(1):e50964.3) Jing X, Yang F, Shao C, Wei K, Xie M, Shen H, Shu Y. Role of hypoxia in cancer therapy by regulating the tumor microenvironment. Mol Cancer. 2019 Nov 11;18(1):157.

Round 2

Reviewer 1 Report

After revision, this work is now acceptable for publication in Nutrients.